# FSPool: Learning Set Representations with Featurewise Sort Pooling

**Yan Zhang**
University of Southampton
Southampton, UK
`yz5n12@ecs.soton.ac.uk`

**Jonathon Hare**
University of Southampton
Southampton, UK
`jsh2@ecs.soton.ac.uk`

**Adam Prügel-Bennett**
University of Southampton
Southampton, UK
`apb@ecs.soton.ac.uk`

## Abstract

Traditional set prediction models can struggle with simple datasets due to an issue we call the responsibility problem. We introduce a pooling method for sets of feature vectors based on sorting features across elements of the set. This can be used to construct a permutation-equivariant auto-encoder that avoids this responsibility problem. On a toy dataset of polygons and a set version of MNIST, we show that such an auto-encoder produces considerably better reconstructions and representations. Replacing the pooling function in existing set encoders with FSPool improves accuracy and convergence speed on a variety of datasets.

## 1 Introduction

Consider the following task: you have a dataset wherein each datapoint is a *set* of 2-d points that form the vertices of a regular polygon, and the goal is to learn an auto-encoder on this dataset. The only variable is the rotation of this polygon around the origin, with the number of points, size, and centre of it fixed. Because the inputs and outputs are sets, this problem has some unique challenges.

**Encoder:** This turns the set of points into a latent space. The order of the elements in the set is irrelevant, so the feature vector the encoder produces should be invariant to permutations of the elements in the set. While there has been recent progress on learning such functions (Zaheer et al., 2017; Qi et al., 2017), they compress a set of any size down to a single feature vector in one step. This can be a significant bottleneck in what these functions can represent efficiently, particularly when relations between elements of the set need to be modeled (Murphy et al., 2019; Zhang et al., 2019b).

**Decoder:** This turns the latent space back into a set. The elements in the target set have an arbitrary order, so a standard reconstruction loss cannot be used naïvely – the decoder would have to somehow output the elements in the same arbitrary order. Methods like those in Achlioptas et al. (2018) therefore use an assignment mechanism to match up elements (section 2), after which a usual reconstruction loss can be computed. Surprisingly, their model is still unable to solve the polygon reconstruction task with close-to-zero reconstruction error, despite the apparent simplicity of the dataset.

In this paper, we introduce a set pooling method for neural networks that addresses both the encoding bottleneck issue and the decoding failure issue. We make the following contributions:

1. We identify the *responsibility problem* (section 3). This is a fundamental issue with existing set prediction models that has not been considered in the literature before, explaining why these models struggle to model even the simple polygon dataset.

2. We introduce FSPool: a differentiable, sorting-based pooling method for variable-size sets (section 4). By using our pooling in the encoder of a set auto-encoder and *inverting the*

*sorting* in the decoder, we can train it with the usual MSE loss for reconstruction *without* the need for an assignment-based loss. This avoids the responsibility problem.

3. We show that our auto-encoder can learn polygon reconstructions with close-to-zero error, which is not possible with existing set auto-encoders (subsection 6.1). This benefit transfers over to a set version of MNIST, where the quality of reconstruction and learned representation is improved (subsection 6.2). In further classification experiments on CLEVR (subsection 6.3) and several graph classification datasets (subsection 6.4), using FSPool in a set encoder improves over many non-trivial baselines. Lastly, we show that combining FSPool with Relation Networks significantly improves over standard Relation Networks in a model that heavily relies on the quality of the representation (subsection 6.5).

## 2 BACKGROUND

The problem with predicting sets is that the output order of the elements is arbitrary, so computing an elementwise mean squared error does not make sense; there is no guarantee that the elements in the target set happen to be in the same order as they were generated. The existing solution around this problem is an assignment-based loss, which assigns each predicted element to its "closest" neighbour in the target set first, after which a traditional pairwise loss can be computed.

We have a predicted set $\hat{Y}$ with feature vectors as elements and a ground-truth set $Y$, and we want to measure how different the two sets are. These sets can be represented as matrices with the feature vectors placed in the columns in some arbitrary order, so $\hat{Y} = [\hat{y}^{(1)}, \ldots, \hat{y}^{(n)}]$ and $Y = [y^{(1)}, \ldots, y^{(n)}]$ with $n$ as the set size (columns) and $d$ as the number of features per element (rows). In this work, we assume that these two sets have the same size. The usual way to produce $\hat{Y}$ is with a multi-layer perceptron (MLP) that has $d \times n$ outputs.

**Linear assignment**   One way to do this assignment is to find a linear assignment that minimises the total loss, which can be solved with the Hungarian algorithm in $O(n^3)$ time. With $\Pi$ as the space of all $n$-length permutations:

$$\mathcal{L}_H(\hat{Y}, Y) = \min_{\pi \in \Pi} \sum_i^n ||\hat{y}^{(i)} - y^{(\pi(i))}||^2 \tag{1}$$

**Chamfer loss**   Alternatively, we can assign each element directly to the closest element in the target set. To ensure that all points in the target set are covered, a term is added to the loss wherein each element in the target set is also assigned to the closest element in the predicted set. This has $O(n^2)$ time complexity and can be run efficiently on GPUs.

$$\mathcal{L}_C(\hat{Y}, Y) = \sum_i \min_j ||\hat{y}^{(i)} - y^{(j)}||^2 + \sum_j \min_i ||\hat{y}^{(i)} - y^{(j)}||^2 \tag{2}$$

Both of these losses are examples of permutation-invariant functions: the loss is the same regardless of how the columns of $Y$ and $\hat{Y}$ are permuted.

## 3 RESPONSIBILITY PROBLEM

It turns out that standard neural networks struggle with modeling symmetries that arise because there are $n!$ different list representations of the same set, which we highlight here with an example. Suppose we want to train an auto-encoder on our polygon dataset and have a square (so a set of 4 points with the x-y coordinates as features) with some arbitrary initial rotation (see Figure 1). Each pair in the 8 outputs of the MLP decoder is *responsible* for producing one of the points in this square. We mark each such pair with a different colour in the figure.

If we rotate the square (top left in figure) by 90 degrees (top right in figure), we simply permute the elements within the set. They are the same set, so they also encode to the same latent representation

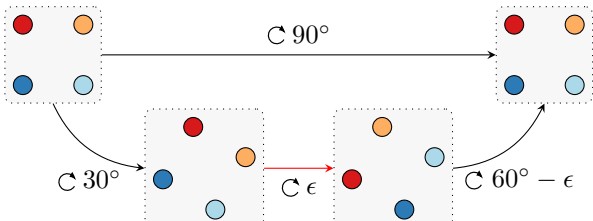

Figure 1: Discontinuity (red arrow) when rotating the set of points. The coloured points denote which output of the network is responsible for which point. In the top path, the set rotated by $90°$ is the same set (exactly the same shape before and after rotation) and encodes to the same feature vector, so the output responsibility (colouring) must be the same too. In this example, after $30°$ and a further small clockwise rotation by $\epsilon$, the point that each output pair is responsible for has to suddenly change.

and decode to the same list representation. This means that each output is still responsible for producing the point at the same position after the rotation, i.e. the dark red output is still responsible for the top left point, the light red output is responsible for the top right point, etc. However, this also means that at some point during that 90 degree rotation (bottom path in figure), *there must exist a discontinuous jump* (red arrow in figure) in how the outputs are assigned. We know that the 90 degree rotation must start and end with the top left point being produced by the dark red output. Thus, we know that there is a rotation where all the outputs must simultaneously change which point they are responsible for, so that completing the rotation results in the top left point being produced by the dark red output. *Even though we change the set continuously, the list representation (MLP or RNN outputs) must change discontinuously.*

This is a challenge for neural networks to learn, since they can typically only model functions without discontinuous jumps. As we increase the number of vertices in the polygon (number of set elements), it must learn an increasing frequency of situations where all the outputs must discontinuously change at once, which becomes very difficult to model. Our experiment in subsection 6.1 confirms this.

This example highlights a more general issue: whenever there are at least two set elements that can be smoothly interchanged, these discontinuities arise. We show this more formally in Appendix A. For example, the set of bounding boxes in object detection can be interchanged in much the same way as the points of our square here. An MLP or RNN that tries to generate these (like in Rezatofighi et al. (2018); Stewart & Andriluka (2016)) must handle which of its outputs is responsible for what element in a discontinuous way. Note that traditional object detectors like Faster R-CNN do not have this responsibility problem, because they do not treat object detection as a proper set prediction task with their anchor-based approach. When the set elements come from a finite domain (often a set of labels) and not $\mathbb{R}^d$, it does not make sense to interpolate set elements. Thus, the responsibility problem does not apply to methods for such problems, for example Welleck et al. (2018); Rezatofighi. et al. (2017).

## 4 FEATUREWISE SORT POOLING

The main idea behind our pooling method is simple: sorting each feature across the elements of the set and performing a weighted sum. The numerical sorting ensures the property of permutation-invariance. The difficulty lies in how to determine the weights for the weighted sum in a way that works for variable-sized sets.

A key insight for auto-encoding is that we can store the permutation that the sorting applies in the encoder and apply the inverse of that permutation in the decoder. This allows the model to restore the arbitrary order of the set element so that it no longer needs an assignment-based loss for training. This avoids the problem in Figure 1, because rotating the square by $90°$ also permutes the outputs of the network accordingly. Thus, there is no longer a discontinuity in the outputs during this rotation. In other words, we make the auto-encoder permutation-*equivariant*: permuting the input set also permutes the neural network's output in the same way.

We describe the model for the simplest case of encoding fixed-size sets in subsection 4.1, extend it to variable-sized sets in subsection 4.2, then discuss how to use this in an auto-encoder in subsection 4.3.

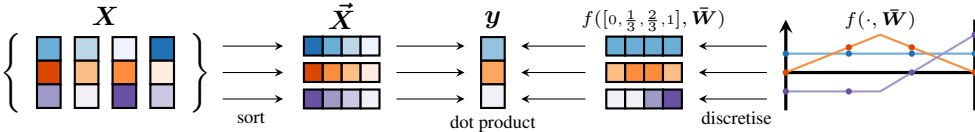

Figure 2: Overview of our FSPOOL model for variable-sized sets. In this example, the weights define piecewise linear functions with two pieces. The four dots on each line correspond to the positions where $f$ is evaluated for a set of size four.

### 4.1 FIXED-SIZE SETS

We are given a set of $n$ feature vectors $\boldsymbol{X} = [\boldsymbol{x}^{(1)}, \ldots, \boldsymbol{x}^{(n)}]$ where each $\boldsymbol{x}^{(i)}$ is a column vector of dimension $d$ placed in some arbitrary order in the columns of $\boldsymbol{X} \in \mathbb{R}^{d \times n}$. From this, the goal is to produce a single feature vector in a way that is invariant to permutation of the columns in the matrix.

We first sort each of the $d$ features across the elements of the set by numerically sorting within the rows of $\boldsymbol{X}$ to obtain the matrix of sorted features $\vec{\boldsymbol{X}}$:

$$\vec{X}_{i,j} = \text{SORT}(\boldsymbol{X}_{i,:})_j \tag{3}$$

where $\boldsymbol{X}_{i,:}$ is the $i$th row of $\boldsymbol{X}$ and $\text{SORT}(\cdot)$ sorts a vector in descending order. While this may appear strange since the columns of $\vec{\boldsymbol{X}}$ no longer correspond to individual elements of the set, there are good reasons for this. A transformation (such as with an MLP) prior to the pooling can ensure that the features being sorted are mostly independent so that little information is lost by treating the features independently. Also, if we were to sort whole elements by one feature, there would be discontinuities whenever two elements swap order. This problem is avoided by our featurewise sorting.

Efficient parallel implementations of SORT are available in Deep Learning frameworks such as PyTorch, which uses a bitonic sort ($O(\log^2 n)$ parallel time, $O(n \log^2 n)$ comparisons). While the permutation that the sorting applies is not differentiable, gradients can still be propagated pathwise according to this permutation in a similar way as for max pooling.

Then, we apply a learnable weight matrix $\boldsymbol{W} \in \mathbb{R}^{d \times n}$ to $\vec{\boldsymbol{X}}$ by elementwise multiplying and summing over the columns (row-wise dot products).

$$y_i = \sum_j^n W_{i,j} \vec{X}_{i,j} \tag{4}$$

$\boldsymbol{y} \in \mathbb{R}^d$ is the final pooled representation of $\vec{\boldsymbol{X}}$. The weight vector allows different weightings of different ranks and is similar in spirit to the parametric version of the gather step in Gather-Excite (Hu et al., 2018). This is a generalisation of both max and sum pooling, since max pooling can be obtained with the weight vector $[1, 0, \ldots, 0]$ and sum pooling can be obtained with the $\mathbf{1}$ vector. Thus, it is also a maximally powerful pooling method for multi-sets (Xu et al., 2019) while being potentially more flexible (Murphy et al., 2019) in what it can represent.

### 4.2 VARIABLE-SIZE SETS

When the size $n$ of sets can vary, our previous weight matrix can no longer have a fixed number of columns. To deal with this, we define a *continuous* version of the weight vector in each row: we use a fixed number of weights to parametrise a piecewise linear function $f : [0, 1] \to \mathbb{R}$, also known as calibrator function (Jaderberg et al., 2015). For a set of size three, this function would be evaluated at 0, 0.5, and 1 to determine the three weights for the weighted sum. For a set of size four, it would be evaluated at 0, 1/3, 2/3, and 1. This decouples the number of columns in the weight matrix from the set size that it processes, which allows it to be used for variable-sized sets.

To parametrise a piecewise linear function $f$, we have a weight vector $\bar{\boldsymbol{w}} \in \mathbb{R}^k$ where $k - 1$ is the number of pieces defined by the $k$ points. With the ratio $r \in [0, 1]$,

$$f(r, \bar{\boldsymbol{w}}) = \sum_{i=1}^{k} \max(0, 1 - |r(k-1) - (i-1)|)\bar{w}_i \tag{5}$$

The $\max(\cdot)$ term selects the two nearest points to $r$ and linearly interpolates them. For example, if $k = 3$, choosing $r \in [0, 0.5]$ interpolates between the first two points in the weight vector with $(1 - 2r)w_1 + 2rw_2$.

We have a different $\bar{\boldsymbol{w}}$ for each of the $d$ features and place them in the rows of a weight matrix $\bar{\boldsymbol{W}} \in \mathbb{R}^{d \times k}$, which no longer depends on $n$. Using these rows with $f$ to determine the weights:

$$y_i = \sum_{j=1}^{n} f(\frac{j-1}{n-1}, \boldsymbol{W}_{i,:})\vec{X}_{i,j} \tag{6}$$

$\boldsymbol{y}$ is now the pooled representation with a potentially varying set size $n$ as input. When $n = k$, this reduces back to Equation 4. For most experiments, we simply set $k = 20$ without tuning it.

## 4.3 AUTO-ENCODER

To create an auto-encoder, we need a decoder that turns the latent space back into a set. Analogously to image auto-encoders, we want this decoder to roughly perform the operations of the encoder in reverse. The FSPool in the encoder has two parts: sorting the features, and pooling the features. Thus, the FSUnpool version should "unpool" the features, and "unsort" the features. For the former, we define an unpooling version of Equation 6 that distributes information from one feature vector to a variable-size list of feature vectors. For the latter, the idea is to store the permutation of the sorting from the encoder and use the inverse of it in the decoder to unsort it. This allows the auto-encoder to restore the original ordering of set elements, which makes it permutation-equivariant.

With $\boldsymbol{y}' \in \mathbb{R}^d$ as the vector to be unpooled, we define the unpooling similarly to Equation 6 as

$$\vec{X}'_{i,j} = f(\frac{j-1}{n-1}, \boldsymbol{W}'_{i,:})y'_i \tag{7}$$

In the non-autoencoder setting, the lack of differentiability of the permutation is not a problem due to the pathwise differentiability. However, in the auto-encoder setting we make use of the permutation in the decoder. While gradients can still be propagated through it, it introduces discontinuities whenever the sorting order in the encoder for a set changes, which we empirically observed to be a problem. To avoid this issue, we need the permutation that the sort produces to be differentiable. To achieve this, we use the recently proposed sorting networks (Grover et al., 2019), which is a continuous relaxation of numerical sorting. This gives us a differentiable approximation of a permutation matrix $\boldsymbol{P}_i \in [0, 1]^{n \times n}, i \in \{1, \ldots, d\}$ for each of the $d$ features, which we can use in the decoder while still keeping the model fully differentiable. It comes with the trade-off of increased computation costs with $O(n^2)$ time and space complexity, so we only use the relaxed sorting in the auto-encoder setting. It is possible to decay the temperature of the relaxed sort throughout training to 0, which allows the more efficient traditional sorting algorithm to be used at inference time.

Lastly, we can use the inverse of the permutation from the encoder to restore the original order.

$$X'_{i,j} = (\vec{\boldsymbol{X}}'_{i,:}\boldsymbol{P}_i^T)_j \tag{8}$$

where $\boldsymbol{P}_i^T$ permutes the elements of the $i$th row in $\vec{X}'$.

Because the permutation is stored and used in the decoder, this makes our auto-encoder similar to a U-net architecture (Long et al., 2015) since it is possible for the network to skip the small latent space. Typically we find that this only starts to become a problem when $d$ is too big, in which case it is possible to only use a subset of the $P_i$ in the decoder to counteract this.

## 5   RELATED WORK

We are proposing a differentiable function that maps a *set* of feature vectors to a single feature vector. This has been studied in many works such as Deep Sets (Zaheer et al., 2017) and PointNet (Qi et al., 2017), with universal approximation theorems being proven. In our notation, the Deep Sets model is $g(\sum_j h(\boldsymbol{X}_{:,j}))$ where $h : \mathbb{R}^d \to \mathbb{R}^p$ and $g : \mathbb{R}^p \to \mathbb{R}^q$. Since this is $O(n)$ in the set size $n$, it is clear that while it may be able to approximate any set function, problems that depend on higher-order interactions between different elements of the set will be difficult to model aside from pure memorisation. This explains the success of relation networks (RN), which simply perform this sum over all *pairs* of elements, and has been extended to higher orders by Murphy et al. (2019). Our work proposes an alternative operator to the sum that is intended to allow some relations between elements to be modeled through the sorting, while not incurring as large of a computational cost as the $O(n^2)$ complexity of RNs.

**Sorting-based set functions**   The use of sorting has often been considered in the set learning literature due to its natural way of ensuring permutation-invariance. The typical approach is to sort elements of the set as units rather than our approach of sorting each feature individually.

For example, the similarly-named SortPooling (Zhang et al., 2018) sorts the elements based on one feature of each element. However, this introduces discontinuities into the optimisation whenever two elements swap positions after the sort. For variable-sized sets, they simply truncate (which again adds discontinuities) or pad the sorted list to a fixed length and process this with a CNN, treating the sorted vectors as a sequence. Similarly, Cangea et al. (2018) and Gao & Ji (2019) truncate to a fixed-size set by computing a score for each element and keeping elements with the top-k scores. In contrast, our pooling handles variable set sizes without discontinuities through the featurewise sort and continuous weight space. Gao & Ji (2019) propose a graph auto-encoder where the decoder use the "inverse" of what the top-k operator does in the encoder, similar to our approach. Instead of numerically sorting, Mena et al. (2018) and Zhang et al. (2019b) *learn* an ordering of set elements instead.

Outside of the set learning literature, rank-based pooling in a convolutional neural network has been used in Shi et al. (2016), where the rank is turned into a weight. Sorting within a single feature vector has been used for modeling more powerful functions under a Lipschitz constraint for Wasserstein GANs (Anil et al., 2018) and improved robustness to adversarial examples (Cisse et al., 2017).

**Set prediction**   Assignment-based losses combined with an MLP or similar are a popular choice for various auto-encoding and generative tasks on point clouds (Fan et al., 2017; Yang et al., 2018; Achlioptas et al., 2018). An interesting alternative approach is to perform the set generation sequentially (Stewart & Andriluka, 2016; Johnson, 2017; You et al., 2018). The difficulty lies in how to turn the set into one or multiple sequences, which these papers try to solve in different ways. Since the initial release of this paper, Zhang et al. (2019a) developed a set prediction method which uses FSPool as a core component and motivate their work by our observations about the responsibility problem. Interestingly, their model uses the gradient of the set encoder, which involves computing the gradient of FSPool; this is closely related to the FSUnpool we proposed.

## 6   EXPERIMENTS

We start with two auto-encoder experiments, then move to tasks where we replace the pooling in an established model with FSPool. Full results can be found in the appendices, experimental details can be found in Appendix H, and we provide our code for reproducibility at [redacted].

### 6.1   ROTATING POLYGONS

We start with our simple dataset of auto-encoding regular polygons (section 3), with each point in a set corresponding to the x-y coordinate of a vertex in that polygon. This dataset is designed to explicitly test whether the responsibility problem occurs in practice. We keep the set size the same within a training run and only vary the rotation. We try this with set sizes of increasing powers of 2.

**Model**   The encoder contains a 2-layer MLP applied to each set element, FSPool, and a 2-layer MLP to produce the latent space. The decoder contains a 2-layer MLP, FSUnpool, and a 2-layer MLP

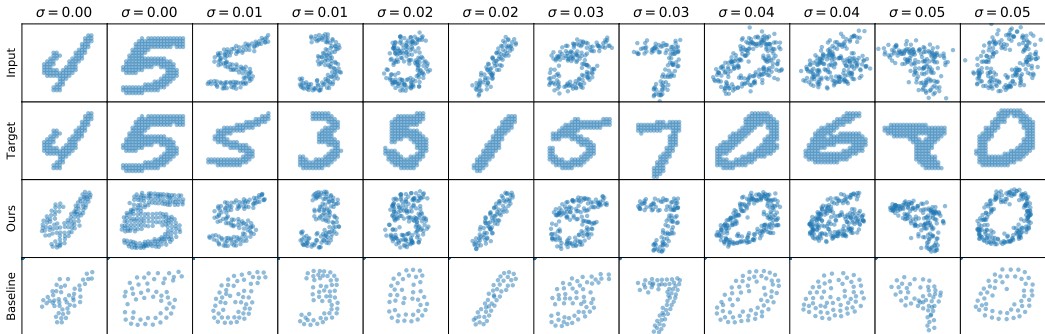

Figure 3: MNIST as point sets with different amounts of Gaussian noise ($\sigma$) and their reconstructions. The baseline uses sum pooling and an MLP decoder, which had the best quantitative results among the baselines. We used the best network for our model ($0.28 \times 10^4$ average Chamfer loss) and the best network for the baseline model ($0.20 \times 10^4$ average Chamfer loss). The examples are not cherry-picked.

applied on each set element. We train this model to minimise the mean squared error. As baseline, we use a model where the decoder has been replaced with an MLP and train it with either the linear assignment or Chamfer loss (equivalent to AE-EMD and AE-CD models in Achlioptas et al. (2018)).

**Results** First, we verified that if the latent space is always zeroed out, the model with FSPool is unable to train, suggesting that the latent space is being used and is necessary. For our training runs with set sizes up to 128, our auto-encoder is able to reconstruct the point set close to perfectly (see Appendix B). Meanwhile, the baseline converges significantly slower with high reconstruction error when the number of points is 8 or fewer and outputs the same set irrespective of input above that, regardless of loss function. Even when significantly increasing the latent size, dimensionality of layers, tweaking the learning rate, and replacing FSPool in the encoder with sum, mean, or max, the baseline trained with the linear assignment or Chamfer loss fails completely at 16 points. We verified that for 4 points, the baseline shows the discontinuous jump behaviour in the outputs as we predict in Figure 1. This experiment highlights the difficulty of learning this simple dataset with traditional approaches due to the responsibility problem, while our model is able to fit this dataset with ease.

## 6.2 NOISY MNIST SETS

Next, we turn to the harder task of auto-encoding MNIST images – turned into sets of points – using a denoising auto-encoder. Each pixel that is above the mean pixel level is considered to be part of the set with its x-y coordinates as feature, scaled to be within the range of [0, 1]. The set size varies between examples and is 133 on average. We add Gaussian noise to the points in the set and use the set without noise as training target for the denoising auto-encoder.

**Model** We use exactly the same architecture as on the polygon dataset. As baseline models, we combine sum/mean/max pooling encoders with MLP/LSTM decoders and train with the Chamfer loss. This closely corresponds to the AE-CD approach (Achlioptas et al., 2018) with the MLP decoder and the model by Stewart & Andriluka (2016) with the LSTM decoder. We tried the approach by Zhang et al. (2019a), but it performs much worse than the other baselines, likely because it requires a bigger encoder (our encoder has $\sim$3000 parameters, their encoder has $\sim$85000 parameters).

**Results** We show example outputs in Figure 3 and the full results in Appendix C. We focus on comparing our FSPool-FSUnpool model against the best baseline, which uses the sum pooling encoder and MLP decoder. In general, our model can reconstruct the digits much better than the baseline, which tends to predict too few points even though it always has 342 (the maximum set size) times 2 outputs available. Occasionally, the baseline also makes big errors such as turning 5s into 8s (first $\sigma = 0.01$ example), which we have not observed with our model.

Table 1: MNIST classification accuracy over 6 runs (different pre-trained networks between runs): mean $\pm$ stdev for $\sigma = 0.05$. Frozen: training with frozen pre-trained auto-encoder weights. Unfrozen: unfrozen auto-encoder weights (fine-tuning). Random init: auto-encoder weights not used.

| | 1 epoch of training | | | 10 epochs of training | | |
|---|---|---|---|---|---|---|
| | Frozen | Unfrozen | Random init | Frozen | Unfrozen | Random init |
| FSPOOL | $82.2\%_{\pm2.1}$ | $86.9\%_{\pm1.3}$ | $84.7\%_{\pm1.9}$ | $84.3\%_{\pm1.8}$ | $91.5\%_{\pm0.5}$ | $91.9\%_{\pm0.5}$ |
| SUM | $76.6\%_{\pm1.3}$ | $68.7\%_{\pm3.5}$ | $30.3\%_{\pm5.6}$ | $79.0\%_{\pm1.0}$ | $77.7\%_{\pm2.3}$ | $72.7\%_{\pm3.4}$ |
| MEAN | $25.7\%_{\pm3.6}$ | $32.2\%_{\pm10.5}$ | $30.1\%_{\pm1.6}$ | $36.8\%_{\pm5.0}$ | $75.0\%_{\pm2.7}$ | $73.0\%_{\pm1.7}$ |
| MAX | $73.6\%_{\pm1.3}$ | $73.0\%_{\pm3.5}$ | $56.1\%_{\pm5.6}$ | $77.3\%_{\pm0.9}$ | $80.4\%_{\pm1.8}$ | $76.9\%_{\pm1.3}$ |

### 6.2.1 CLASSIFICATION

Instead of auto-encoding MNIST sets, we can also classify them. We use the same dataset and replace the set decoder in our model and the baseline with a 2-layer MLP classifier. We consider three variants: using the trained auto-encoder weights for the encoder and freezing them, not freezing them (finetuning), and training all weights from random initialisation. This tests how informative the learned representations of the pre-trained auto-encoder and the encoder are.

**Results**   We show our results for $\sigma = 0.05$ in Table 1. Results for $\sigma = 0.00$ and 100 epochs are shown in Appendix D. Even though our model can store information in the permutation that skips the latent space, our latent space contains more information to correctly classify a set, even when the weights are fixed. Our model with fixed encoder weights already performs better after 1 epoch of training than the baseline models with unfrozen weights after 10 epochs of training. This shows the benefit of the FSPool-FSUnpool auto-encoder to the representation. When allowing the encoder weights to change (Unfrozen and Random init), our results again improve significantly over the baselines. Interestingly, switching the relaxed sort to the unrelaxed sort in our model when using the fixed auto-encoder weights does not hurt accuracy. Training the FSPool model takes 45 seconds per epoch on a GTX 1080 GPU, only slightly more than the baselines with 37 seconds per epoch.

### 6.3 CLEVR

CLEVR (Johnson, 2017) is a visual question answering dataset where the task is to classify an answer to a question about an image. The images show scenes of 3D objects with different attributes, and the task is to answer reasoning questions such as "what size is the sphere that is left of the green thing". Since we are interested in sets, we use this dataset with the ground-truth state description – the set of objects (maximum size 10) and their attributes – as input instead of an image of the rendered scene.

**Model**   For this dataset, we compare against relation networks (RN) (Santoro et al., 2017) – explicitly modeling all pairwise relations – Janossy pooling (Murphy et al., 2019), and regular pooling functions. While the original RN paper reports a result of 96.4% for this dataset, we use a tuned implementation by Messina et al. (2018) with 2.6% better accuracy. For our model, we modify this to not operate on pairwise relations and replace the existing sum pooling with FSPool. We use the same hyperparameters for our model as the strong RN baseline without further tuning them.

**Results**   Over 10 runs, Table 2 shows that our FSPool model reaches the best accuracy and also reaches the listed accuracy milestones in fewer epochs than all baselines. The difference in accuracy is statistically significant (two-tailed t-tests against sum, mean, RN, all with $p \approx 0.01$). Also, FSPool reaches 99% accuracy in 5.3 h, while the fastest baseline, mean pooling, reaches the same accuracy in 6.2 h. Surprisingly, RNs do not provide any benefit here, despite the hyperparameters being explicitly tuned for the RN model. We show some of the functions $f(\cdot, \bar{W})$ that FSPool has learned in Appendix E. These confirm that FSPool uses more complex functions than just sums or maximums, which allow it to capture more information about the set than other pooling functions.

Table 2: CLEVR results over 10 runs: mean $\pm$ stdev of accuracy after 350 epochs, epochs to reach an accuracy milestone, and wall time required with a 1080 Ti GPU. * averages over only 8 runs because 2 runs did not reach 99%. MAC (Hudson & Manning, 2018) is a model specifically designed for CLEVR and the state-of-the-art for *image inputs* and without program supervision.

| | | Epochs to reach accuracy | | | Time for |
|---|---|---|---|---|---|
| Model | Accuracy | 98.00% | 98.50% | 99.00% | 350 epochs |
| FSPOOL | $\mathbf{99.27\%}_{\pm 0.18}$ | $\mathbf{141}_{\pm 5}$ | $\mathbf{166}_{\pm 16}$ | $\mathbf{209}_{\pm 33}$ | 8.8 h |
| RN | $98.98\%_{\pm 0.25}$ | $144_{\pm 6}$ | $189_{\pm 29}$ | $*268_{\pm 46}$ | 15.5 h |
| JANOSSY | $97.00\%_{\pm 0.54}$ | – | – | – | 11.5 h |
| SUM | $99.05\%_{\pm 0.17}$ | $146_{\pm 13}$ | $191_{\pm 40}$ | $281_{\pm 56}$ | 8.0 h |
| MEAN | $98.96\%_{\pm 0.27}$ | $169_{\pm 6}$ | $225_{\pm 31}$ | $273_{\pm 33}$ | 8.0 h |
| MAX | $96.99\%_{\pm 0.26}$ | – | – | – | 8.0 h |
| MAC | 99.0 % | – | – | – | – |

### 6.4 GRAPH CLASSIFICATION

We perform a large number of experiments on various graph classification datasets from the TU repository (Kersting et al., 2016): 4 graph datasets from bioinformatics (for example with the graph encoding the structure of a molecule) and 5 datasets from social networks (for example with the graph encoding connectivity between people who worked with each other). The task is to classify the whole graph into one of multiple classes such as positive or negative drug response.

**Model**   We use the state-of-the-art graph neural network GIN (Xu et al., 2019) as baseline. This involves a series of graph convolutions (which includes aggregation of features from each node's set of neighbours into the node), a readout (which aggregates the set of all nodes into one feature vector), and a classification with an MLP. We replace the usual sum or mean pooling readout with FSPool $k = 5$ for our model. We repeat 10-fold cross-validation on each dataset 10 times and use the same hyperparameter ranges as Xu et al. (2019) for our model and the GIN baseline.

**Results**   We show the results in Appendix F. On 6 out of 9 datasets, FSPool achieves better test accuracy. On a different 6 datasets, it converges to the best validation accuracy faster. A Wilcoxon signed-rank test shows that the difference in accuracy to the standard GIN has $p \approx 0.07$ ($W = 7$) and the difference in convergence speed has $p \approx 0.11$ ($W = 9$). Keep in mind that just because the results have $p > 0.05$, it does not mean that the results are invalid.

### 6.5 CLEVR WITH DEEP SET PREDICTION NETWORKS

Zhang et al. (2019a) build on the ideas in this paper to develop a model that can predict sets from an image. Their model requires from a set encoder that the more similar two set inputs are, the more similar their representations should be. This is harder than classification, because different inputs (of the same class) should no longer map to the same representation. In this experiment, we quantify the benefit of the RN + FSPool set encoder they used. We use their experimental set-up and replace FSPool with sum (this gives the normal RN model) or max pooling. We train this on CLEVR to predict the set of bounding boxes or the state description (this was the input in subsection 6.3).

**Results**   Appendix G shows that for both bounding box and state prediction models, the RN encoder using FSPool is much better than sum or max. This shows that it is possible to improve on standard Relation Networks simply by replacing the sum with FSPool when the task is challenging enough.

## 7 DISCUSSION

In this paper, we identified the responsibility problem with existing approaches for predicting sets and introduced FSPool, which provides a way around this issue in auto-encoders. In experiments on two datasets of point clouds, we showed that this results in much better reconstructions. We

believe that this is an important step towards set prediction tasks with more complex set elements. However, because our decoder uses information from the encoder, it is not easily possible to turn it into a generative set model, which is the main limitation of our approach. Still, we find that using the auto-encoder to obtain better representations and pre-trained weights can be beneficial by itself. Our insights about the responsibility problem have already been successfully used to create a model without the limitations of our auto-encoder (Zhang et al., 2019a).

In classification experiments, we also showed that simply replacing the pooling function in an existing model with FSPool can give us better results and faster convergence. We showed that FSPool consistently learns better set representations at a relatively small computational cost, leading to improved results in the downstream task. Our model thus has immediate applications in various types of set models that have traditionally used sum or max pooling. It would be useful to theoretically characterise what types of relations are more easily expressed by FSPool through an analysis like in Murphy et al. (2019). This may result in further insights into how to learn better set representations efficiently.

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

## A    FORMAL RESPONSIBILITY PROBLEM

The following theorem is a more formal treatment of the responsibility problem resulting in discontinuities.

**Theorem 1.** *For any set function $f : \mathcal{S}_n^d \to \mathbb{R}^{d \times n}$ ($d \geq 2$, $n \geq 2$, $\mathcal{S}_n^d$ is the set of all sets of size $n$ with elements in $\mathbb{R}^d$) from a set of points $\boldsymbol{S} = \{\boldsymbol{x}_1, \boldsymbol{x}_2, \ldots, \boldsymbol{x}_n\}$ to a list representation of that set $\boldsymbol{L} = [\boldsymbol{x}_{\sigma(1)}, \boldsymbol{x}_{\sigma(2)}, \ldots, \boldsymbol{x}_{\sigma(n)}]$ with some fixed permutation $\sigma \in \Pi$, there will be a discontinuity in $f$: there exists an $\varepsilon > 0$ such that for all $\delta > 0$, there exist two sets $\boldsymbol{S}_1$ and $\boldsymbol{S}_2$ where:*

$$d_s(\boldsymbol{S}_1, \boldsymbol{S}_2) < \delta \quad \text{and} \quad d_l(f(\boldsymbol{S}_1), f(\boldsymbol{S}_2)) \geq \varepsilon. \tag{9}$$

*$d_s$ is a measure of the distance between two sets (e.g. Chamfer loss) and $d_l$ is the sum of Euclidean distances ($d_l(\boldsymbol{A}, \boldsymbol{B}) = \sum_j \|\boldsymbol{a}_j - \boldsymbol{b}_j\|_2$).*

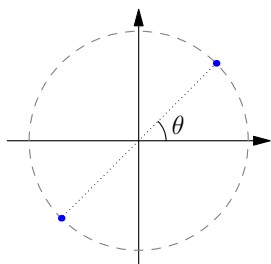

Figure 4: Example of the set with two points.

*Proof.* We prove the theorem by considering mappings from a set of two points in two dimensions. For larger sets or sets with more dimensions, we can isolate two points and two dimensions and ignore the remaining points and dimensions.

Let us consider the set of two points $\boldsymbol{S}(\theta) = \left\{ \begin{bmatrix} -\cos(\theta) \\ -\sin(\theta) \end{bmatrix}, \begin{bmatrix} \cos(\theta) \\ \sin(\theta) \end{bmatrix} \right\}$ (see Figure 4). This is mapped to a list $\boldsymbol{L}(\theta) = f(\boldsymbol{S}(\theta))$. Without loss of generality, we can assume that our list representation for $\theta = 0$ is $\boldsymbol{L}(0) = \begin{bmatrix} -\cos(0) & \cos(0) \\ -\sin(0) & \sin(0) \end{bmatrix} = \begin{bmatrix} -1 & 1 \\ 0 & 0 \end{bmatrix}$. Since the order of set elements is irrelevant and $f$ is a (permutation-invariant) set function, $\boldsymbol{S}(\pi) = \boldsymbol{S}(0)$ and therefore $\boldsymbol{L}(\pi) = \boldsymbol{L}(0) = \begin{bmatrix} -1 & 1 \\ 0 & 0 \end{bmatrix}$. This implies that for at least one value of $\theta = \theta^*$, there is a change in responsibility such that for $\theta \leq \theta^*$, the list representation will be $\boldsymbol{L}_1(\theta) = \begin{bmatrix} -\cos(\theta) & \cos(\theta) \\ -\sin(\theta) & \sin(\theta) \end{bmatrix}$ while for $\theta > \theta^*$, the list representation will be $\boldsymbol{L}_2(\theta) = \begin{bmatrix} \cos(\theta) & -\cos(\theta) \\ \sin(\theta) & -\sin(\theta) \end{bmatrix}$ in order to satisfy $\boldsymbol{L}(\pi) = \boldsymbol{L}(0)$. For any $\theta$, $d_l(\boldsymbol{L}_1(\theta), \boldsymbol{L}_2(\theta)) = 4$.

Let $\varepsilon = 3.9$ and $\delta$ be given. We can find a sufficiently small $\alpha > 0$ so that $d_s(\boldsymbol{S}(\theta^*), \boldsymbol{S}(\theta^* + \alpha)) < \delta$ and $d_l(\boldsymbol{L}(\theta^*), \boldsymbol{L}(\theta^* + \alpha)) > \varepsilon$. $\quad\square$

The reason why this does not apply to our method is that rather than choosing a fixed $\sigma$ for the list representation, the permutation-equivariance (instead of the invariance of set functions) allows our model to have $\boldsymbol{L}(\pi) \neq \boldsymbol{L}(0)$.

## B    POLYGONS

**Results**    In Table 3, Table 4, and Table 5, we show the results of various model and training loss combinations. We include a random baseline that outputs a polygon with the correct size and centre, but random rotation.

These show that FSPool with the direct MSE training loss is clearly better than the baseline with either linear assignment or Chamfer loss on all the evaluation metrics. When the set size is 16 or greater, the other combinations only perform as well as the random baseline because they output the same constant set regardless of input.

Table 3: Direct mean squared error (in hundredths) on Polygon dataset with different number of points in the set. Lower is better.

| Set size | 2 | 4 | 8 | 16 | 32 | 64 |
|---|---|---|---|---|---|---|
| FSPool | 0.000 | 0.001 | 0.000 | 0.000 | 0.000 | 0.0001 |
| Random | 100.323 | 100.134 | 99.367 | 99.951 | 99.438 | 99.523 |

Table 4: Chamfer loss (in hundredths) on Polygon dataset with different number of points in the set. Lower is better.

| Set size | 2 | 4 | 8 | 16 | 32 | 64 |
|---|---|---|---|---|---|---|
| FSPool | 0.001 | 0.001 | 0.001 | 0.000 | 0.001 | 0.002 |
| MLP + Chamfer | 1.189 | 1.771 | 0.274 | 1.272 | 0.316 | 0.085 |
| MLP + Hungarian | 1.517 | 0.400 | 0.251 | 1.266 | 0.326 | 0.081 |
| Random | 72.848 | 19.866 | 5.112 | 1.271 | 0.322 | 0.081 |

Table 5: Linear assignment loss (in hundredths) on Polygon dataset with different number of points in the set. Lower is better.

| Set size | 2 | 4 | 8 | 16 | 32 | 64 |
|---|---|---|---|---|---|---|
| FSPool | 0.000 | 0.001 | 0.000 | 0.000 | 0.000 | 0.001 |
| MLP + Chamfer | 0.595 | 0.885 | 0.137 | 0.641 | 0.160 | 0.285 |
| MLP + Hungarian | 0.758 | 0.200 | 0.126 | 0.634 | 0.163 | 0.040 |
| Random | 36.424 | 9.933 | 2.556 | 0.635 | 0.161 | 0.041 |

## C   MNIST RECONSTRUCTION

**Results**   We show the results for the default MNIST setting in Table 6. Interestingly, the sum pooling baseline has a lower Chamfer reconstruction error than our model, despite the example outputs in Figure 3 looking clearly worse. This demonstrates a weakness of the Chamfer loss. Our model avoids this weakness by being trained with a normal MSE loss (with the cost of a potentially higher Chamfer loss), which is not possible with the baselines. The sum pooling baseline has a better test Chamfer loss because it is trained to minimise it, but it is also solving an easier task, since it does not need to distinguish padding from non-padding elements.

The main reason for this difference comes from the shortcoming of the Chamfer loss in distinguishing sets with duplicates or near-duplicates. For example, the Chamfer loss between $[1, 1.001, 9]$ and $[1, 9, 9.001]$ is close to 0. Most points in an MNIST set are quite close to many other points and there are many duplicate padding elements, so this problem with the Chamfer loss is certainly present on MNIST. That is why minimising MSE can lead to different results with higher Chamfer loss than minimising Chamfer loss directly, even though the qualitative results seem worse for the latter.

We can make the comparison between our model and the baselines more similar by forcing the models to predict an additional "mask feature" for each set element. This takes the value 1 when the point is present (non-padding element) and 0 (padding element) when not. This setting is useful for tasks where the predicted set size matters, as it allows points at the coordinates (0, 0) to be distinguished from padding elements. These padding elements are necessary for efficient minibatch-wise training.

The results of this variant are shown in Table 7. Now, our model is clearly better: even though our auto-encoder minimises an MSE loss, the test Chamfer loss is also much better than all the baselines. Having to predict this additional mask feature does not affect our model predictions much because our model structure lets our model "know" which elements are padding elements, while this is much more challenging for the baselines.

Table 6: Test Chamfer loss (in 10 000ths) for MNIST for different input noise levels $\sigma$ over 6 runs. Lower is better.

| Noise $\sigma$ | 0.00 | 0.01 | 0.02 | 0.03 | 0.04 | 0.05 |
|---|---|---|---|---|---|---|
| FSPOOL + FSUNPOOL | $0.42_{\pm0.06}$ | $0.34_{\pm0.05}$ | $0.36_{\pm0.02}$ | $0.38_{\pm0.03}$ | $0.41_{\pm0.00}$ | $0.44_{\pm0.01}$ |
| SUM + MLP | $\mathbf{0.30}_{\pm0.04}$ | $\mathbf{0.28}_{\pm0.03}$ | $\mathbf{0.28}_{\pm0.03}$ | $\mathbf{0.28}_{\pm0.03}$ | $\mathbf{0.27}_{\pm0.01}$ | $\mathbf{0.31}_{\pm0.04}$ |
| SUM + RNN | $0.76_{\pm0.46}$ | $0.58_{\pm0.06}$ | $0.57_{\pm0.09}$ | $0.54_{\pm0.11}$ | $0.64_{\pm0.13}$ | $0.78_{\pm0.39}$ |
| MAX + MLP | $1.29_{\pm0.23}$ | $1.37_{\pm0.28}$ | $1.23_{\pm0.16}$ | $1.74_{\pm0.32}$ | $1.27_{\pm0.19}$ | $1.43_{\pm0.30}$ |
| MEAN + MLP | $1.41_{\pm0.12}$ | $1.22_{\pm0.18}$ | $1.33_{\pm0.29}$ | $1.25_{\pm0.09}$ | $1.31_{\pm0.15}$ | $1.49_{\pm0.31}$ |

Table 7: Test Chamfer loss (in 10 000ths) for MNIST with additional mask features (see description in Appendix C) on every element for different input noise levels $\sigma$ over 6 runs. Lower is better.

| Noise $\sigma$ | 0.00 | 0.01 | 0.02 | 0.03 | 0.04 | 0.05 |
|---|---|---|---|---|---|---|
| FSPOOL + FSUNPOOL | $\mathbf{0.28}_{\pm0.03}$ | $\mathbf{0.21}_{\pm0.01}$ | $\mathbf{0.25}_{\pm0.03}$ | $\mathbf{0.25}_{\pm0.01}$ | $\mathbf{0.27}_{\pm0.01}$ | $\mathbf{0.30}_{\pm0.01}$ |
| SUM + MLP | $0.87_{\pm0.43}$ | $0.90_{\pm0.39}$ | $0.61_{\pm0.40}$ | $0.99_{\pm0.84}$ | $0.63_{\pm0.27}$ | $0.61_{\pm0.24}$ |
| SUM + RNN | $0.58_{\pm0.13}$ | $0.69_{\pm0.16}$ | $0.60_{\pm0.18}$ | $1.35_{\pm1.53}$ | $0.73_{\pm0.12}$ | $0.63_{\pm0.13}$ |
| MAX + MLP | $5.91_{\pm3.10}$ | $4.78_{\pm3.05}$ | $7.10_{\pm2.40}$ | $5.05_{\pm2.87}$ | $5.85_{\pm3.11}$ | $4.57_{\pm2.08}$ |
| MEAN + MLP | $5.92_{\pm3.10}$ | $4.55_{\pm3.17}$ | $6.84_{\pm2.84}$ | $7.11_{\pm2.39}$ | $3.04_{\pm0.78}$ | $6.34_{\pm2.53}$ |

Table 8: Classification accuracy (mean $\pm$ stdev) on MNIST $\sigma = 0.00$ over 6 runs.

| | 1 epoch of training | | | 10 epochs of training | | |
|---|---|---|---|---|---|---|
| | Frozen | Unfrozen | Random init | Frozen | Unfrozen | Random init |
| FSPOOL | $\mathbf{86.3\%}_{\pm1.6}$ | $\mathbf{92.3\%}_{\pm1.1}$ | $\mathbf{90.5\%}_{\pm1.2}$ | $\mathbf{88.2\%}_{\pm1.4}$ | $\mathbf{96.0\%}_{\pm0.3}$ | $\mathbf{96.1\%}_{\pm0.3}$ |
| SUM | $82.3\%_{\pm1.2}$ | $77.9\%_{\pm3.4}$ | $35.3\%_{\pm8.3}$ | $85.0\%_{\pm0.8}$ | $84.2\%_{\pm2.5}$ | $78.4\%_{\pm3.9}$ |
| MEAN | $27.0\%_{\pm3.3}$ | $43.5\%_{\pm7.1}$ | $31.2\%_{\pm1.0}$ | $42.0\%_{\pm7.7}$ | $76.7\%_{\pm2.6}$ | $77.2\%_{\pm2.2}$ |
| MAX | $82.0\%_{\pm1.8}$ | $84.1\%_{\pm1.4}$ | $62.9\%_{\pm3.5}$ | $86.8\%_{\pm0.9}$ | $91.9\%_{\pm1.3}$ | $87.7\%_{\pm1.2}$ |

Table 9: Classification accuracy (mean $\pm$ stdev) on MNIST for 100 epochs over 6 runs.

| | $\sigma = 0.05$, 100 epochs | | | $\sigma = 0.00$, 100 epochs | | |
|---|---|---|---|---|---|---|
| | Frozen | Unfrozen | Random init | Frozen | Unfrozen | Random init |
| FSPOOL | $\mathbf{84.9\%}_{\pm1.7}$ | $\mathbf{93.9\%}_{\pm0.4}$ | $\mathbf{94.0\%}_{\pm0.3}$ | $88.6\%_{\pm1.6}$ | $\mathbf{97.4\%}_{\pm0.3}$ | $\mathbf{97.5\%}_{\pm0.3}$ |
| SUM | $79.8\%_{\pm1.0}$ | $85.3\%_{\pm1.1}$ | $83.1\%_{\pm1.9}$ | $85.6\%_{\pm0.9}$ | $89.5\%_{\pm2.5}$ | $88.3\%_{\pm1.4}$ |
| MEAN | $48.2\%_{\pm6.9}$ | $86.5\%_{\pm0.8}$ | $84.1\%_{\pm2.3}$ | $57.0\%_{\pm7.7}$ | $90.3\%_{\pm1.3}$ | $91.1\%_{\pm0.8}$ |
| MAX | $78.8\%_{\pm0.8}$ | $84.7\%_{\pm1.0}$ | $84.6\%_{\pm0.9}$ | $\mathbf{89.2\%}_{\pm0.8}$ | $95.3\%_{\pm0.7}$ | $95.1\%_{\pm1.5}$ |

# D  MNIST CLASSIFICATION

**Results**   Table 8 and Table 9 show the results for $\sigma = 0.00$ and for 100 epochs for both $\sigma = 0.05$ and $\sigma = 0.00$ respectively. Note that these are based on pre-trained models from the default MNIST setting without mask feature. Like before, the FSPool-based models are consistently superior to all the baselines. Note that while (Qi et al., 2017) report an accuracy of ∼99% on a similar set version of MNIST, our model uses noisy sets as input and is much smaller and simpler: we have 3820 parameters, while their model has 1.6 million parameters. Our model also does not use dropout, batch norm, a branching network architecture, and a stepped learning rate schedule. When we try to match their model size, our accuracies for $\sigma = 0.00$ increase to ∼99% as well.

# E    CLEVR

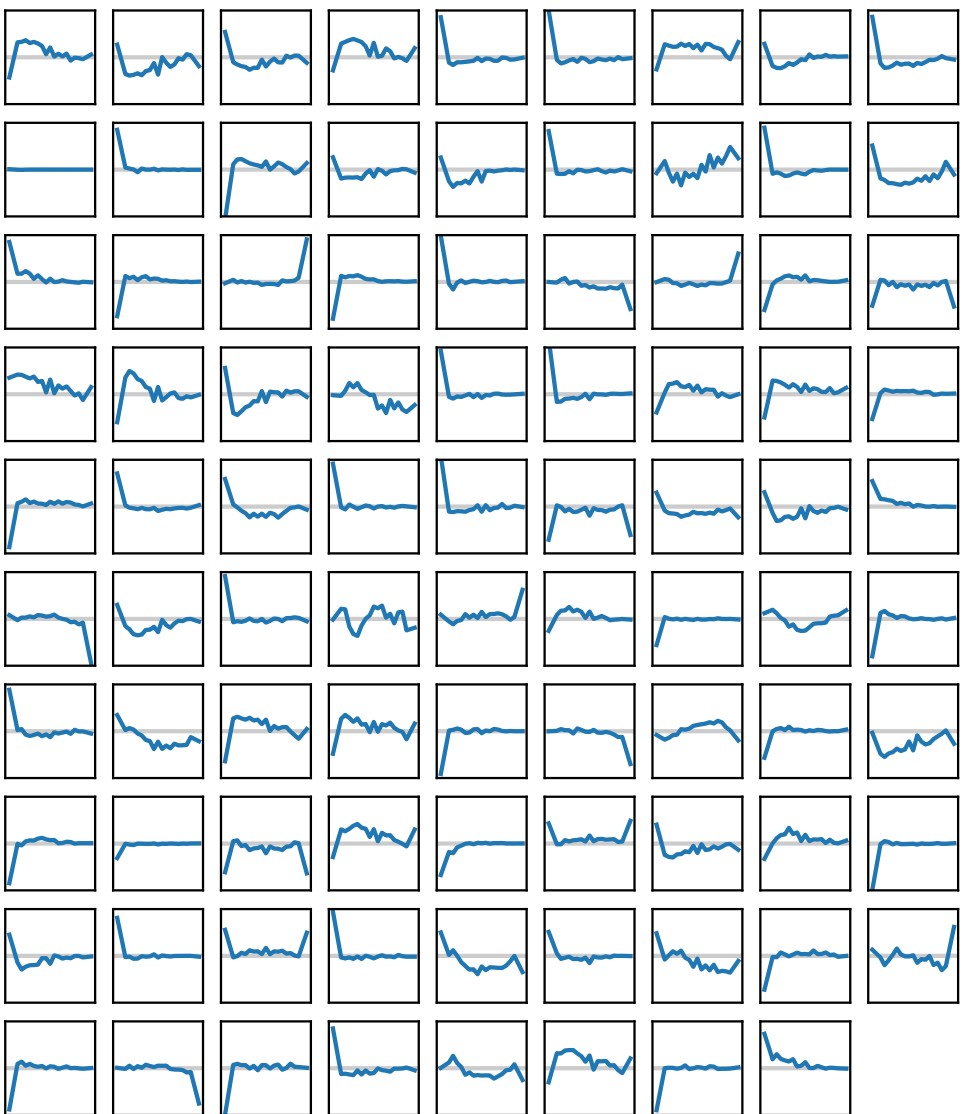

Figure 5: Shapes of piecewise linear functions learned by the FSPool model on CLEVR. These show $r \in [0, 1]$ on the x-axis and $f(r, \bar{w})$ on the y-axis for a particular $\bar{w}$ of a fully-trained model. A common shape among these functions are variants of max pooling: close to 0 weight for most ranks and a large non-zero weight on either the maximum or the minimum value, for example in row 2 column 2. There are many functions that simple maximums or sums can *not* easily represent, such as a variant of max pooling with the values slightly below the max receiving a weight of the opposite sign (see row 1 column 1) or the shape in the penultimate row column 5. The functions shown here may have a stronger tendency towards 0 values than normal due to the use of weight decay on CLEVR.

# F    GRAPH CLASSIFICATION

**Experimental setup**    The datasets and node features used are the same as in GIN; we did not cherry-pick them. Because the social network datasets are purely structural without node features, a

Table 10: Cross-validation classification results (%) on various commonly-used graph classification datasets, with the mean cross-validation accuracy averaged over 10 repeats and sample standard deviations ($\pm$). Hyperparameters of entries marked with * are known to be selected based on test accuracy instead of validation accuracy, so results are likely not comparable to other existing approaches that were (hopefully) selected based on validation accuracy. Our results were selected based on validation accuracy.

| Social Network | IMDB-B | IMDB-M | RDT-B | RDT-M5K | COLLAB |
|---|---|---|---|---|---|
| Num. graphs | 1000 | 1500 | 2000 | 5000 | 5000 |
| Num. classes | 2 | 3 | 2 | 5 | 3 |
| Avg. nodes | 19.8 | 13.0 | 429.6 | 508.5 | 74.5 |
| Max. nodes | 136 | 89 | 3063 | 2012 | 492 |
| DCNN (Atwood & Towsley, 2016) | 49.1 | 33.5 | – | – | 52.1 |
| PATCHY-SAN (Niepert et al., 2016) | 71.0 $\pm2.3$ | 45.2 $\pm2.8$ | 86.3 $\pm1.6$ | 49.1 $\pm0.7$ | 72.6 $\pm2.2$ |
| SORTPOOL (Zhang et al., 2018) | 70.0 $\pm0.9$ | 47.8 $\pm0.9$ | – | – | 73.8 $\pm0.5$ |
| DIFFPOOL (Ying et al., 2018) | – | – | – | – | 75.5 |
| WL* (Xu et al., 2019) | 73.8 | 50.9 | 81.0 | 52.5 | 78.9 |
| GIN-BASE* (Xu et al., 2019) | 75.1 | 52.3 | 92.4 | 57.5 | 80.2 |
| GIN-FSPOOL | **72.1** $\pm2.0$ | **49.9** $\pm1.7$ | **89.1** $\pm1.2$ | **51.8** $\pm0.9$ | 80.0 $\pm0.4$ |
| - epochs | 95 $\pm70$ | **27** $\pm23$ | **124** $\pm64$ | **66** $\pm31$ | **124** $\pm56$ |
| GIN-BASE | 71.3 $\pm1.2$ | 48.8 $\pm1.7$ | 84.8 $\pm1.7$ | 48.1 $\pm2.0$ | **80.3** $\pm0.4$ |
| - epochs | **83** $\pm73$ | 57 $\pm59$ | 156 $\pm58$ | 211 $\pm27$ | 204 $\pm26$ |

| Bioinformatics | MUTAG | PROTEINS | PTC | NCI1 |
|---|---|---|---|---|
| Num. graphs | 188 | 1113 | 344 | 4110 |
| Num. classes | 2 | 2 | 2 | 2 |
| Avg. nodes | 17.9 | 39.1 | 25.5 | 29.8 |
| Max. nodes | 28 | 620 | 109 | 111 |
| PK (Neumann et al., 2016) | 76.0 $\pm2.7$ | 73.7 $\pm0.7$ | 59.5 $\pm2.4$ | 82.5 $\pm0.5$ |
| DCNN (Atwood & Towsley, 2016) | 67.0 | 61.3 | 56.6 | 62.6 |
| PATCHY-SAN (Niepert et al., 2016) | 92.6 $\pm4.2$ | 75.9 $\pm2.8$ | 60.0 $\pm4.8$ | 78.6 $\pm1.9$ |
| SORTPOOL (Zhang et al., 2018) | 85.8 $\pm1.7$ | 75.5 $\pm0.9$ | 58.6 $\pm2.5$ | 74.4 $\pm0.5$ |
| DIFFPOOL (Ying et al., 2018) | – | 76.3 | – | – |
| WL (Shervashidze et al., 2011) | 84.1 $\pm1.9$ | 74.7 $\pm0.5$ | 58.0 $\pm2.5$ | 85.5 $\pm0.5$ |
| WL* (Xu et al., 2019) | 90.4 | 75.0 | 59.9 | 86.0 |
| GIN-BASE* (Xu et al., 2019) | 89.4 | 76.2 | 64.6 | 82.7 |
| GIN-FSPOOL | **85.9** $\pm2.4$ | **73.8** $\pm0.9$ | 59.3 $\pm1.8$ | 79.2 $\pm0.6$ |
| - epochs | 299 $\pm91$ | **69** $\pm23$ | 214 $\pm110$ | **361** $\pm54$ |
| GIN-BASE | 85.0 $\pm1.5$ | 73.2 $\pm1.2$ | **59.9** $\pm2.4$ | **79.4** $\pm0.6$ |
| - epochs | **244** $\pm95$ | 160 $\pm123$ | **202** $\pm100$ | 412 $\pm55$ |

constant 1 feature is used on the RDT datasets and the one-hot-encoded node degree is used on the other social network datasets. The hyperparameter sweep is done based on best validation accuracy for each fold in the cross-validation individually and over the same combinations as specified in GIN.

Note that in GIN, hyperparameters are selected based on best *test* accuracy. This is a problem, because they consider the number of epochs a hyperparameter when accuracies tend to significantly vary between individual epochs. For example, our average result on the PROTEINS dataset would change from 73.8% to 77.1% if we were to select based on best test accuracy, which would be better than their 76.2%.

While we initially also used $k = 20$ in FSPool for this experiment, we found that $k = 5$ was consistently an improvement. The $k = 20$ model was still better than the baseline on average by a smaller margin.

**Results** We show our results of GIN-FSPool and the GIN baseline averaged over 10 repeats in Table 10. On the majority of datasets, FSPool has slightly better accuracies than the strong baseline and consistently takes fewer epochs to reach its highest validation accuracy. On the two RDT datasets, this improvement is large. Interestingly, these are the two datasets where the number of nodes to be pooled is by far the largest with an average of 400+ nodes per graph, compared to the next largest COLLAB with an average of 75 nodes. This is perhaps evidence that FSPool is helping to avoid the bottleneck problem of pooling a large set of feature vectors to a single feature vector.

We emphasise that the main comparison to be made is between the GIN-Base and the GIN-FSPool model, since that is the only comparison where the only factor of difference is the pooling method. When comparing against other models, the network architecture, training hyperparameters, and evaluation methodology can differ significantly.

Keep in mind that while GIN-Base looks much worse than the original GIN-Base*, the difference is that our implementation has hyperparameters properly selected by validation accuracy, while GIN-Base* selected them by test accuracy. If we were to select based on test accuracy, our implementation frequently outperforms their results. Also, they only performed a single run of 10-fold cross-validation.

## G   DEEP SET PREDICTION NETWORKS

Table 11: Average Precision (AP, mean $\pm$ stdev) for different intersection-over-union thresholds of the predicted bounding boxes over 6 runs. DSPN-RN-FSPool results are taken from Zhang et al. (2019a).

| Model | $AP_{50}$ | $AP_{90}$ | $AP_{95}$ | $AP_{98}$ | $AP_{99}$ |
|---|---|---|---|---|---|
| DSPN-RN-FSPOOL (10 iters) | $98.8_{\pm0.3}$ | $94.3_{\pm1.5}$ | $85.7_{\pm3.0}$ | $\mathbf{34.5}_{\pm5.7}$ | $\mathbf{2.9}_{\pm1.2}$ |
| DSPN-RN-FSPOOL (20 iters) | $\mathbf{99.8}_{\pm0.0}$ | $\mathbf{98.7}_{\pm1.1}$ | $\mathbf{86.2}_{\pm7.2}$ | $24.3_{\pm8.0}$ | $1.4_{\pm0.9}$ |
| DSPN-RN-FSPOOL (30 iters) | $\mathbf{99.8}_{\pm0.1}$ | $96.7_{\pm2.4}$ | $75.5_{\pm12.3}$ | $17.4_{\pm7.7}$ | $0.9_{\pm0.7}$ |
| DSPN-RN-SUM (10 iters) | $88.3_{\pm3.7}$ | $43.4_{\pm14.4}$ | $10.0_{\pm7.4}$ | $0.1_{\pm0.1}$ | $0.0_{\pm0.0}$ |
| DSPN-RN-SUM (20 iters) | $87.2_{\pm3.0}$ | $42.9_{\pm11.9}$ | $5.7_{\pm3.5}$ | $0.0_{\pm0.0}$ | $0.0_{\pm0.0}$ |
| DSPN-RN-SUM (30 iters) | $79.0_{\pm11.9}$ | $32.5_{\pm12.4}$ | $3.4_{\pm2.2}$ | $0.0_{\pm0.0}$ | $0.0_{\pm0.0}$ |
| DSPN-RN-MAX (10 iters) | $68.0_{\pm4.3}$ | $4.0_{\pm2.2}$ | $0.1_{\pm0.1}$ | $0.0_{\pm0.0}$ | $0.0_{\pm0.0}$ |
| DSPN-RN-MAX (20 iters) | $66.6_{\pm4.5}$ | $3.3_{\pm1.8}$ | $0.1_{\pm0.0}$ | $0.0_{\pm0.0}$ | $0.0_{\pm0.0}$ |
| DSPN-RN-MAX (30 iters) | $64.1_{\pm5.0}$ | $2.3_{\pm1.1}$ | $0.0_{\pm0.0}$ | $0.0_{\pm0.0}$ | $0.0_{\pm0.0}$ |

Table 12: Average Precision (AP, mean $\pm$ stdev) for different distance thresholds of the predicted state descriptions over 6 runs. DSPN-RN-FSPool results are taken from Zhang et al. (2019a).

| Model | $AP_{\infty}$ | $AP_1$ | $AP_{0.5}$ | $AP_{0.25}$ | $AP_{0.125}$ |
|---|---|---|---|---|---|
| DSPN-RN-FSPOOL (10 iters) | $72.8_{\pm2.3}$ | $59.2_{\pm2.8}$ | $39.0_{\pm4.4}$ | $12.4_{\pm2.5}$ | $1.3_{\pm0.4}$ |
| DSPN-RN-FSPOOL (20 iters) | $84.0_{\pm4.5}$ | $80.0_{\pm4.9}$ | $\mathbf{57.0}_{\pm12.1}$ | $\mathbf{16.6}_{\pm9.0}$ | $\mathbf{1.6}_{\pm0.9}$ |
| DSPN-RN-FSPOOL (30 iters) | $\mathbf{85.2}_{\pm4.8}$ | $\mathbf{81.1}_{\pm5.2}$ | $47.4_{\pm17.6}$ | $10.8_{\pm9.0}$ | $0.6_{\pm0.7}$ |
| DSPN-RN-SUM (10 iters) | $44.6_{\pm3.8}$ | $21.9_{\pm4.8}$ | $7.1_{\pm2.7}$ | $1.0_{\pm0.5}$ | $0.0_{\pm0.0}$ |
| DSPN-RN-SUM (20 iters) | $39.6_{\pm5.4}$ | $15.2_{\pm6.4}$ | $3.0_{\pm2.2}$ | $0.3_{\pm0.3}$ | $0.0_{\pm0.0}$ |
| DSPN-RN-SUM (30 iters) | $30.2_{\pm9.2}$ | $7.1_{\pm3.8}$ | $0.9_{\pm0.8}$ | $0.1_{\pm0.1}$ | $0.0_{\pm0.0}$ |
| DSPN-RN-MAX (10 iters) | $3.0_{\pm0.2}$ | $0.9_{\pm0.1}$ | $0.5_{\pm0.2}$ | $0.1_{\pm0.1}$ | $0.0_{\pm0.0}$ |
| DSPN-RN-MAX (20 iters) | $3.1_{\pm0.1}$ | $1.2_{\pm0.1}$ | $0.8_{\pm0.2}$ | $0.3_{\pm0.2}$ | $0.0_{\pm0.0}$ |
| DSPN-RN-MAX (30 iters) | $3.1_{\pm0.1}$ | $1.2_{\pm0.1}$ | $0.9_{\pm0.2}$ | $0.3_{\pm0.2}$ | $0.0_{\pm0.0}$ |

**Results** Table 11 and Table 12 show that the FSPool-based RN encoder is much better than any of the baselines. The representation of DSPN-RN-FSPool is good enough that iterating the DSPN algorithm for more steps than the model was trained with can benefit the prediction, while for the baselines it generally just worsens.

This is especially apparent for the harder dataset of state prediction, where more information has to be compressed into the latent space.

# H  EXPERIMENTAL DETAILS

We provide the code to reproduce all experiments at [redacted].

For almost all experiments, we used FSPool and the unpooling version of it with $k = 20$. We guessed this value without tuning, and we did not observe any major differences when we tried to change this on CLEVR to $k = 5$ and $k = 40$. $\bar{W}$ can be initialised in different ways, such as by sampling from a standard Gaussian. However, for the purposes of starting the model as similarly as possible to the sum pooling baseline on CLEVR and on the graph classification datasets, we initialise $\bar{W}$ to a matrix of all 1s on them.

## H.1  POLYGONS

The polygons are centred on 0 with a radius of 1. The points in the set are randomly permuted to remove any ordering in the set from the generation process that a model that is not permutation-invariant or permutation-equivariant could exploit. We use a batch size of 16 for all three models and train it for 10240 steps. We use the Adam optimiser (Kingma & Ba, 2015) with 0.001 learning rate and their suggested values for the other optimiser parameters (PyTorch defaults). Weights of linear and convolutional layers are initialised as suggested in Glorot & Bengio (2010). The size of every hidden layer is set to 16 and the latent space is set to 1 (it should only need to store the rotation as latent variable). We have also tried much hidden and latent space sizes of 128 when we tried to get better results for the baselines.

## H.2  MNIST RECONSTRUCTION

We train on the training set of MNIST for 10 epochs and the shown results come from the test set of MNIST. For an image, the coordinate of a pixel is included if the pixel is above the mean pixel level of 0.1307 (with pixel levels ranging 0–1). Again, the order of the points are randomised. We did not include results of the linear assignment loss because we did not get the model to converge to results of similar quality to the direct MSE loss or Chamfer loss, and training time took too long ($> 1$ day) in order to find better parameters.

The latent space is increased from 1 to 16 and the size of the hidden layers is increased from 16 to 32. All other hyperparameters are the the same as for the Polygons dataset.

## H.3  CLEVR

The architecture and hyperparameters come from the third-party open-source implementation available at `https://github.com/mesnico/RelationNetworks-CLEVR`. For the RN baseline, the set is first expanded into the set of all pairs by concatenating the 2 feature vectors of the pair for all pairs of elements in the set. For the Janossy Pooling baseline, we use the model configuration from Murphy et al. (2019) that appeared best in their experiments, which uses $\pi$-SGD with an LSTM that has $|h|$ as neighbourhood size.

The question representation coming from the 256-unit LSTM, processing the question tokens in reverse with each token embedded into 32 dimensions, is concatenated to all elements in the set. Each element of this new set is first processed by a 4-layer MLP with 512 neurons in each layer and ReLU activations. The set of feature vectors is pooled with a pooling method like sum and the output of this is processed with a 3-layer MLP (hidden sizes 512, 1024, and number of answer classes) with ReLU activations. A dropout rate of 0.05 is applied before the last layer of this MLP. Adam is used with a starting learning rate of 0.000005, which doubles every 20 epochs until the maximum learning rate of 0.0005 is reached. Weight decay of 0.0001 is applied. The model is trained for 350 epochs.

Table 13: Average of best hyperparameters over 10 repeats.

| | IMDB-B | IMDB-M | RDT-B | RDT-M5K | COLLAB | MUTAG | PROTEINS | PTC | NCI1 |
|---|---|---|---|---|---|---|---|---|---|
| GIN-FSPOOL | | | | | | | | | |
| *- dimensionality* | 64.0 | 64.0 | 64.0 | 64.0 | 64.0 | 28.8 | 19.2 | 28.8 | 30.4 |
| *- batch size* | 66.0 | 100 | 45.6 | 32.0 | 86.4 | 89.6 | 60.8 | 41.6 | 128 |
| *- dropout* | 0.25 | 0.15 | 0.35 | 0.10 | 0.40 | 0.15 | 0.35 | 0.20 | 0.50 |
| GIN-BASE | | | | | | | | | |
| *- dimensionality* | 64.0 | 64.0 | 64.0 | 64.0 | 64.0 | 27.2 | 20.8 | 25.6 | 28.8 |
| *- batch size* | 86.4 | 93.2 | 72.8 | 100 | 100 | 70.4 | 60.8 | 60.8 | 128 |
| *- dropout* | 0.30 | 0.15 | 0.25 | 0.45 | 0.40 | 0.25 | 0.45 | 0.20 | 0.35 |

## H.4    GRAPH CLASSIFICATION

The GIN architecture starts with 5 sequential blocks of graph convolutions. Each block starts with summing the feature vector of each node's neighbours into the node's own feature vector. Then, an MLP is applied to the feature vectors of all the nodes individually. The details of this MLP were somewhat unclear in Xu et al. (2019) and we chose Linear-ReLU-BN-Linear-ReLU-BN in the end. We tried Linear-BN-ReLU-Linear-BN-ReLU as well, which gave us slightly worse validation results for both the baseline and the FSPool version. The outputs of each of the 5 blocks are concatenated and pooled, either with a sum for the social network datasets, mean for the social network datasets (this is as specified in GIN), or with FSPool for both types of datasets. This is followed by BN-Linear-ReLU-Dropout-Linear as classifier with a softmax output and cross-entropy loss. We used the torch-geometric library (Fey et al., 2018) to implement this model.

The starting learning rate for Adam is 0.01 and is reduced every 50 epochs. Weights are initialised as suggested in Glorot & Bengio (2010). The hyperparameters to choose from are: dropout ratio $\in \{0, 0.5\}$, batch size $\in \{32, 128\}$, if bioinformatics dataset hidden sizes of all layers $\in \{16, 32\}$ and 500 epochs, if social network dataset the hidden size is 64 and 250 epochs. Due to GPU memory limitations we used a batch size of 100 instead of 128 for social network datasets. The best hyperparameters are selected based on best average validation accuracy across the 10-fold cross-validation, where one of the 9 training folds is used as validation set each time. In other words, within one 10-fold cross-validation run the hyperparameters used for the test set are the same, while across the 10 repeats of this with different seeds the best hyperparameters may differ.

## H.5    DEEP SET PREDICTION NETWORKS

The architecture and hyperparameters come from the third-party open-source implementation available at `https://github.com/Cyanogenoid/dspn`. The only thing we change from this is replacing the pooling in the RN. All other hyperparameters are kept the same.

The input image is encoded with a ResNet-34 with two additional convolutional layers with 512 filters and stride two to obtain a feature vector for the image. This feature vector is decoded into a set using the DSPN algorithm, which requires encoding an intermediate set with the set encoder and performing gradient descent on it. This set encoder creates all pairs of sets like in normal RNs, processes each pair with a 2-layer MLP with 512 neurons with one ReLU activation in the middle, then pools this into a feature vector. The intermediate set is updated with the gradient 10 times in training, but can be iterated a different amount in evaluation. The model is trained to minimise the linear assignment loss with the Adam optimiser for 100 epochs using a learning rate of 0.0003.

