# OpenReview forum: "FSPool: Learning Set Representations with Featurewise Sort Pooling"
_ICLR.cc/2020/Conference — Accept (Poster)_

### Official Review · AnonReviewer1 · 2019-10-24
**Official Blind Review #1**

**Rating:** 6

**Review:**

The authors point out an interesting problem that the responsibility between target and input is not continuous and propose an FSPool method to alleviate this problem. The method is simple and easy to implement.

However, there are still some questions the authors don't answer.
Firstly, the authors point out that this issue also exists in some tasks like object detection, but I have no idea how to apply this method in object detection to fix this issue.

Secondly, the authors make a comparison between FSPool and sum pool, average pool, and max pool, however, what about the weighted sum pool? I think it's most similar to FSPool. Maybe I misunderstand here. But please respond at my concerns and I'll change the score accordingly.

**Experience Assessment:**

I have read many papers in this area.

**Review Assessment: Checking Correctness Of Derivations And Theory:**

I assessed the sensibility of the derivations and theory.

**Review Assessment: Checking Correctness Of Experiments:**

I assessed the sensibility of the experiments.

**Review Assessment: Thoroughness In Paper Reading:**

I read the paper at least twice and used my best judgement in assessing the paper.

---

> ### Author Response · Authors · 2019-11-07
> **Rebuttal**
>
> 1. When we say that the responsibility problem exists in tasks like object detection, we specifically mean the use of MLPs or RNNs to predict each bounding box, rather than the more common approach of models like Faster R-CNN which use anchors. This set-based object detection approach with MLPs/RNNs is something that for example Stewart & Andriluka (2016) have done, which we mention in the related works section.
>
> The anchor-based approach of Faster R-CNN and similar does not have the responsibility issue we describe, because it does not treat object detection as a proper set prediction task in the first place.
>
> We will add a sentence to make this distinction clearer.
>
>
> 2. An important aspect when working with sets is the property of permutation-invariance: changing the order of the set elements (which is arbitrary) should not change the output of the model. FSPool, sum, average, and max pool are all permutation-invariant, but a weighted sum is not because it gives different weights to different arbitrary positions in the set. A weighted sum is therefore not very suited for pooling sets. It is also not clear what weights should be used for variable-size sets.
>
> That is why we do not compare against it. From the Janossy pooling results in Table 2, you can see how a model that is not permutation-invariant (an LSTM in this case) results in worse performance here.

---

### Official Review · AnonReviewer3 · 2019-10-25
**Official Blind Review #3**

**Rating:** 8

**Review:**

This paper presents a new approach to representing sets of inputs and a strategy for set auto-encoders. Multiple experiments demonstrate that the new approach performs better than baseline models from recent literature around set representations. The forward encoding strategy is simple enough for a practitioner to implement, and likely to perform better (in terms of training time/gradient flow; if not also test metrics) than existing sum, min, max, mean pooling strategies. It is not substantially more expensive.

The central trick here is to use a per-feature sort across the set as the representation of the set. In cases of arbitrarily sized sets, the authors provide and use a piecewise linear interpolation strategy, and suggest other possibilities (splines, etc), to induce a uniform representation shape regardless of the input set.

The decoder uses a similar trick to expand a latent value back to input-set-size, and then leverages the argsort from the input to re-permute the expanded set. They point out that this helps to avoid discontinuities otherwise caused by the 'responsibility problem', i.e. which feature is responsible to describe which input element[s].

The experiments seem to cover a lot of ground:
- toy polygon dataset demonstrated to be hard for existing sota in set representations
- sets of points from mnist images
- graph classification (competitive with a recent graph convolution approach)
- integrate with Relation Nets for deep set prediction

The authors acknowledge (sec 7) the limitation imposed by requiring the input argsort (and size) at the decoder, but point out that even as a representational pre-training or regularization, this strategy can help to improve set prediction strategies not subject to the same constraint (like RN, as in 6.5).

I found the work to be well presented, the experiments to be strong, and I think it will be interesting to the community. Recommend accepting.

FYI: It seems this work has been previously shared, presumably on arxiv, judging from some amount of back-and-forth citations, building upon Zhang et al 2019.

**Experience Assessment:**

I have read many papers in this area.

**Review Assessment: Checking Correctness Of Derivations And Theory:**

I assessed the sensibility of the derivations and theory.

**Review Assessment: Checking Correctness Of Experiments:**

I assessed the sensibility of the experiments.

**Review Assessment: Thoroughness In Paper Reading:**

I read the paper at least twice and used my best judgement in assessing the paper.

---

> ### Author Response · Authors · 2019-11-07
> **Rebuttal**
>
> Let us know if you have any questions.

---

### Official Review · AnonReviewer4 · 2019-11-03
**Official Blind Review #4**

**Rating:** 8

**Review:**

This paper proposes to make permutation-equivariant set autoencoders, by introducing a permutation invariant encoder (based on feature-wise sorting) and at the same time an 'inverse' operation (that undoes the sorting) in the decoder. The achieved equivariance allows the autoencoders to be trained end-2-end without a matching-based loss (like Chamfer) which I think is a very neat result. Also, to address potentially variable-sized sets as inputs/outputs, the authors propose to use a calibrator function that effectively 'samples' the produced features on the set's elements in fixed intervals, that are auto-adaptable in the (variable) set size.
The results shown are well placed in the current literature and form an important novelty that should have non-trivial impact in the sub-field. Also, the experiments done are well executed, with ample variability in the nature of the datasets used and I expect them to also be easily reproducible (given the authors' provided code).

Some points I would appreciate to see an improvement:
a. The intro is rough (please see minor-specifics at places that you can improve the exposition).
b. The Figure1 is not easy to read. The 90degree rotation results in the same set, but as a 'pictorial' image, obviously not.
c. The "responsibility problem" is already very well explained in the Zhang19a. I would appreciate to tone-down in this paper, the "discovery" of it as a main contribution.
d. Experiments 6.2. it appears that the FSPool/unpool model is better *only* when the mask features are been considered. What are these mask-features? Why do they matter?
e. The way you describe the responsibility problem (discontinuity) is very hand-wavy. It would be nice to explicitly it write it in rigorous math.
f. Why using the relaxation of Grover et al. helped you to avoid the discontinuity that would be otherwise introduced via standard sorting? (I am not familiar with their exact relaxation, but intuitively, their method been a good proxy for sorting, should suffer from it as well).


Explicit Minor Comments on writing:
(all in introduction)
-4rth line: "this" -> this problem
-"Methods like by" -> Methods like those in
-"In this paper, we introduce a set pooling method for neural networks that addresses both issues" -> which issues? the encoder's collapse and the decoder's inneficiency in matching? Please explain.
-"good baselines" -> sophisticated/non-trivial baselines

Appendix. Table 9, max-pool at \sigma=0, seems to be the best (please use boldface to indicate it).

**Experience Assessment:**

I have published one or two papers in this area.

**Review Assessment: Checking Correctness Of Derivations And Theory:**

I assessed the sensibility of the derivations and theory.

**Review Assessment: Checking Correctness Of Experiments:**

I carefully checked the experiments.

**Review Assessment: Thoroughness In Paper Reading:**

I read the paper at least twice and used my best judgement in assessing the paper.

---

> ### Author Response · Authors · 2019-11-07
> **Rebuttal**
>
> a. Thank you for the improvements to wording in the introduction, we will include them all in the revision. With "both issues", we mean the bottleneck problem of the encoder and the responsibility problem of the decoder.
> Well spotted on the single max result in Table 9 that is slightly better than FSPool, we will of course bold the correct entry in the revision.
>
>
> b. This disconnect between what should happen pictorially (90 degree rotation should rotate outputs by 90 degrees) and what actually happens because it is a set (after a 90 degree rotation it is still the same set, so the output is the same too) is exactly what we want to show in this figure. While we explain this a bit in the accompanying text, we will add a sentence to the caption that elaborates on this. If you have any other ideas on what would make the figure easier to understand, let us know.
>
>
> c. Zhang19a cite the pre-print of our paper for the responsibility problem and placed their description of it in their background section. Their description of the responsibility problem argues that intuitively, it *could* be a problem, while our description shows that it specifically introduces discontinuities which *are* a problem (which also becomes clear from our polygon experiment). In terms of pre-print timeline, our paper introduced the responsibility problem first, Zhang19a then cited it and gave an alternative example of it. So, we maintain that it is a main contribution of our paper.
>
>
> d. The mask feature is a single feature that is concatenated to each element of the set (i.e. each set element in MNIST now has the 3 dimensions x, y, and mask), taking the value 1 for normal elements of the set and 0 for padding elements. This is to explicitly distinguish normal set elements from padding elements, which the model could otherwise interpret as points at coordinates (0, 0). Padding is necessary for minibatch training.
> We briefly describe this Appendix B, but we will add a sentence to the table 7 caption to make this clearer.
>
> This explicit distinction between padding and non-padding elements seems to be enough to make the task hard enough for the baseline to start struggling. We mention this in the Appendix B text, which you might have missed.
>
> To elaborate on the comment in Appendix B that the Chamfer loss has some weaknesses: FSPool-FSUnpool model minimises MSE, while the baseline minimises the Chamfer loss. These two losses can sometimes be "misaligned", which would explain why our Chamfer loss is worse (since the baseline explicitly minimises Chamfer loss), yet most of our outputs in Figure 3 look qualitatively better. One shortcoming of the Chamfer loss is that it does not do well with duplicate and near duplicate elements: the loss between [1, 1.001, 9] and [1, 9, 9.001] is close to 0. Most points in an MNIST set are quite close to many other points and there are many duplicate padding elements, so this problem with the Chamfer loss certainly applies to MNIST. This difference between Chamfer loss and MSE can make a model that is trained with MSE appear worse when evaluated with Chamfer when comparing it to a model that is trained with the Chamferloss.
>
> If you think that this explanation will help a reader understand our results better, we can add this in the revision.
>
> (We realise that it can be a bit hard to tell what the input and target look like in Figure 3 when trying to compare results qualitatively, so we will change the figure to not plot input and target on top of each other in the revision.)
>
>
> e. We will add an appendix that states the responsibility problem in more precise mathematical terms.
>
>
> f. The problem is not the sorting step, but the permutation that the sorting produces. A normal sort produces a "hard" permutation (each entry before the sort is assigned to exactly one position after the sort), while Grover's method produces a "soft" permutation (each entry is assigned mostly to one position after the sort, but also a bit to the nearby ones).
>
> While the hard permutation isn't a problem when just using the encoder (gradient can just be propagated pathwise), in the auto-encoder the permutation that the sort produces is used in FSUnpool. To train the auto-encoder, we therefore need the permutation itself to be differentiable, which can be done for the soft permutation matrix that Grover's method produces, but not the hard permutation of the normal sort.
>
> We can change the text after equation 7 to make this point a bit more clearly.

---

> > ### Comment · AnonReviewer4 · 2019-11-07
> > **Thank you for the rebuttal**
> >
> > With your rebuttal you have addressed my main points. I will increase my score to reflect this. Thank you.

---

### Author Response · Authors · 2019-11-13
**Changes in revision**

We have just uploaded a revision of our paper with the following changes:


# R4
- We added an appendix (Appendix A) with a more formal treatment of the responsibility problem.
- We fixed all the minor things that were pointed out.
- We added a sentence to the caption of Figure 1 that explains why the 90 degree rotation of the set does not rotate the colours.
- The table 7 caption now refers the reader to the description of what the mask feature is in the same appendix (previously Appendix B, now Appendix C). We made the description of what the mask feature is and why it makes sense more explicit. We also included our description of the difference between minimising Chamfer and MSE.
- We now state more clearly after equation 7 that the reason for using the sorting relaxation is to get a differentiable permutation matrix.

# R1
- We added a sentence to the end of section 3 explaining why the anchor-based object detectors do not have the responsibility problem.

# General
- We updated Figure 3 to show input and target more clearly. The caption now also contains information about the Chamfer losses of the shown models (baseline model has lower Chamfer loss, yet looks worse).
- We made the caption of Table 8 and 9 slightly less confusing.

---

### Decision · Program_Chairs · 2019-12-19

**Decision:**

Accept (Poster)

**Comment:**

Overall, this paper got strong scores from the reviewers (2 accepts and 1 weak accept).  The paper proposes to address the responsibility problem, enabling encoding and decoding sets without worrying about permutations.  This is achieved using permutation-equivariant set autoencoders and an 'inverse' operation that undoes the sorting in the decoder.  The reviewers all agreed that the paper makes a meaningful contribution and should be accepted.  Some concerns regarding clarity of exposition were initially raised but were addressed during the rebuttal period.  I recommend that the paper be accepted.